# Sex Differences in Cancer Incidence Rates by Race and Ethnicity: Results from the Surveillance, Epidemiology, and End Results (SEER) Registry (2000–2019)

**DOI:** 10.3390/cancers16050989

**Published:** 2024-02-29

**Authors:** Sararat Tosakoon, Wayne R. Lawrence, Meredith S. Shiels, Sarah S. Jackson

**Affiliations:** Division of Cancer Epidemiology and Genetics, National Cancer Institute, Rockville, MD 20850, USA

**Keywords:** sex differences, cancer epidemiology, cancer registries, SEER

## Abstract

**Simple Summary:**

While males exhibit higher incidence and mortality rates for most non-sex-specific cancers compared to females, the reasons behind these differences remain unclear. By analyzing the male-to-female rate ratios (MF IRRs) of non-sex-specific cancers by race and ethnicity, this research aims to shed light on whether sex-based differences are rooted in intrinsic biological variations or environmental exposures. Similarities of MF IRRs across race and ethnicity may suggest underlying biological variations that drive sex differences, while variations in MF IRRs may point to external factors driving these incidence differences. The results may help us understand the etiology of non-sex-specific cancers and the complex interplay between biology and environment in cancer development.

**Abstract:**

Men have 2–3 times the rate of most non-sex-specific cancers compared to women, but whether this is due to differences in biological or environmental factors remains poorly understood. This study investigated sex differences in cancer incidence by race and ethnicity. Cancer incidence data from the Surveillance, Epidemiology, and End Result (SEER) program (2000–2019) were used to calculate male-to-female incidence rate ratios (MF IRRs) for each cancer site, stratified by race and ethnicity, and age-standardized to the 2000 U.S. population for individuals ages ≥ 20 years. Among 49 cancer sites, 44 showed male predominance (MF IRR > 1), with seven inconsistencies across race and ethnicity, including cancers of the lip, tongue, hypopharynx, retroperitoneum, larynx, pleura cancers, and Kaposi sarcoma. Four cancers exhibited a female predominance (MF IRR < 1), with only gallbladder and anus cancers varying by race and ethnicity. The MF IRRs for cancer of the cranial nerves and other nervous system malignancies showed no sex differences and were consistent (MF IRR = 1) across race and ethnicity. The MF IRRs for most cancers were consistent across race and ethnicity, implying that biological etiologies are driving the observed sex difference. The lack of MF IRR variability by race and ethnicity suggests a minimal impact of environmental exposure on sex differences in cancer incidence. Further research is needed to identify biological drivers of sex differences in cancer etiology.

## 1. Introduction

The incidence and mortality for most non-sex-specific cancers are greater among males than females [1,2,3,4]. Most of these cancers exhibit 2 to 3 times higher male-to-female incidence rate ratios (MF IRRs), with the exceptions of thyroid, gallbladder, and anus cancers, which have a female predominance [1]. However, whether sex differences in cancer incidence are due to intrinsic biological factors (e.g., sex hormones, chromosomes, and gene expression) or environmental exposures remains poorly understood. 

Intrinsic biological factors may explain sex differences in cancer if the MF IRR remains consistent across geography and culture. Previous research using global cancer incidence data found that the observed sex difference of the 13 of 35 studied cancer sites could not be explained by environmental risk factors (e.g., smoking, alcohol consumption, or occupational exposures) [5]. The MF IRR for these cancers did not vary by geographical region or over time, implying that potential differences in cultural, behavioral, and environmental factors by sex did not explain the observed sex difference in cancer incidence [5]. Subsequent research that accounted for potential cancer risk factors concluded that only a minor fraction of the sex-related variability in cancers predominantly affecting males could be linked to differences in risk behaviors and environmental exposures between men and women [6]. 

Exposures to cancer risk factors may vary by sex among different racial and ethnic groups. For example, the prevalence of smoking over 1997 to 2018 for non-Hispanic White individuals is similar between men (22.3%) and women (19.7%), while there are fairly stark differences in smoking prevalence among Hispanic/Latino men (18.4%) and women (9.7%) [7]. Similarly, there are few differences in the prevalence of obesity seen between men and women among non-Hispanic White (White) and Hispanic/Latino (Latino) adults, while a larger proportion of non-Hispanic Black (Black) women (57.9%) are obese compared to their male counterparts (40.4%) [8]. If the observed sex differences can be explained by these risk factors, the MF IRR should vary by race and ethnicity. 

This study investigated sex differences in cancer incidence rates by race and ethnicity to provide a greater understanding of whether intrinsic biological sex differences or behavioral and environmental exposures drive sex differences in cancer incidence rates. Similarities in the MF IRRs of cancers across race and ethnicity are a possible indication of intrinsic biological differences. In contrast, differences in the MF IRRs by race and ethnicity could point to extrinsic etiologies related to exposure differences between men and women within racial and ethnic groups.

## 2. Materials and Methods

Cancer incidence data from the Surveillance, Epidemiology, and End Result (SEER) program from 2000 to 2019 were used to estimate MF IRRs, after being age-adjusted to the 2000 U.S. standard population. SEER includes cancer registries from the following geographic areas within the U.S.: Connecticut, Massachusetts, New Jersey, New York, Kentucky, Louisiana, Atlanta, Rural Georgia, Greater Georgia, Texas, Iowa, Illinois, Hawaii, New Mexico, Seattle–Puget Sound, Utah, San Fransico–Oakland, Los Angeles, Greater California, Idaho, Arizona, Alaska, and the Cherokee Nation. The SEER program covers 47.9% of the U.S. total population, including 42.0% of the White population, 44.7% of the Black population, 70.7% of the Asian population, 70.3% of the Native Hawaiian/Pacific Islander population, 66.3% of the Latino population, and 49.0% of the non-Hispanic American Indian/Alaska Native (AIAN) population [9]. Cancer sites and morphologies and histology subtypes were defined by the International Classification of Disease for Oncology, Third Edition (ICD-O-3) and the World Health Organization 2008 (Appendix A).

MF IRRs and 95% confidence intervals (CIs) were calculated for each cancer site, and selected histologic subtypes were stratified by race and ethnicity using SEER*stat (version 8.4.0.1) [10]. The racial and ethnic groups included were White, Black, non-Hispanic Asian Pacific Islander (API), Latino, and AIAN. AIAN individuals were restricted to those in a purchase/referred care delivery area (PRCDA), where ascertainment of AIAN race is known to be greater. It should be noted that only 3% of the API population is Native Hawaiian or Pacific Islander; therefore, the results for this aggregated group are driven by Asian individuals. Individuals with unknown race and ethnicity were excluded from the analysis. Individuals aged 20 years and older were included and categorized into 10-year age groups (20–29, 30–39, 40–49, 50–59, 60–69, 70–79, and 80+ years) for each race and ethnicity when examining MF IRRs of cancer sites and histologic subtypes.

## 3. Results

Table 1 shows the age-adjusted MF IRRs of all primary cancer sites, stratified by race and ethnicity (male and female incidence rates per 100,000 for each cancer by race and ethnicity are shown in Appendix A). Forty-four of forty-nine cancer sites exhibited a male bias with MF IRRs > 1, of which, all but seven were consistent across race and ethnicity. There were four cancer sites (gallbladder, thyroid, anal, and peritoneum) with MF IRRs < 1, indicating a female predominance, and all but gallbladder and anus cancers were consistent across racial and ethnic groups. Of the 49 cancer sites examined, only cranial nerves and other nervous system malignancies exhibited no sex bias (MF IRR = 1), and this was consistent across race and ethnicity. 

The seven cancer sites with a male bias that varied by race and ethnicity were cancers of the lip, tongue, hypopharynx, retroperitoneum, larynx, pleura, and Kaposi sarcoma. For lip cancer the sex bias was smaller for API individuals (MF IRR: 1.43, 95% CI: 1.05, 1.96) than for the other groups, where the MF IRR was >2.6. A similar pattern was seen for tongue cancer, where the sex bias was smaller for API individuals (MF IRR: 1.74, 95% CI: 1.62, 1.87) than the other racial and ethnic groups, where the MF IRR was >2. For cancer of the hypopharynx, the male incidence rate was 4–6 times that of females in most groups, except among Latino individuals where the MF IRR was 8.18 (95% CI: 6.68, 10.10). For cancer of the retroperitoneum, there was a male bias only for White (MF IRR: 1.27, 95% CI: 1.19, 1.35) and Latino (1.14; 0.99, 1.32) individuals, while there was a female bias observed among Black individuals (0.78; 0.65, 0.93), and no sex bias was shown for API and AIAN individuals. For larynx cancer, the male incidence was 4–5 times that of females among White, Black, and AIAN individuals, while the rate was 7–8 times higher among API (MF IRR: 7.69; 95% CI: 7.05, 8.39) and Latino (8.68; 7.54, 10.03) men. For pleura cancer, the male incidence rate was 2–5 times higher than women among White, Latino, and AIAN individuals, but no sex bias was seen in Black (MF IRR: 0.91; 95% CI 0.37, 2.07) or API (0.76; 0.27, 2.01) individuals. The male bias was pronounced for Kaposi sarcoma among White, Black, and Latino individuals, having an MF IRR ≈ 13 while a smaller sex bias was seen among API (MF IRR: 7.96; 95% CI: 6.78, 9.40) and AIAN (6.33; 2.57, 20.83) individuals.

Cancers of the thyroid and peritoneum exhibited a consistent female bias across race and ethnicity, while cancers of the gallbladder and anus varied by race and ethnicity. Gallbladder cancer exhibited a female bias with females having 1.4–1.6 times the risk of men, but this bias was most pronounced among Latina females who had twice the risk as males (MF IRR: 0.46; 95% CI: 0.43, 0.50). Anus cancer exhibited a female bias across most racial and ethnic groups except for Black individuals, where the incidence rate was higher among males (MF IRR: 1.24; 95% CI: 1.15, 1.33). 

Among histological subtypes shown in Table 2 (male and female incidence rates are shown in Appendix A), all but one cancer site had a male bias, which was mostly consistent across racial and ethnic groups with two notable exceptions. The incidence rate for esophagus adenocarcinoma among males was 6–7 times higher than females among White, API, and Latino individuals, while the incidence rate was 4 times higher for males than females among Black and AIAN individuals. The MF IRR for lung and bronchus large cell carcinoma was largely consistent across race and ethnicity at approximately 2. Lung and bronchus small cell carcinoma were the only histology that showed a female bias, though this was not consistent across race and ethnicity, where the MF IRR for API individuals was 1.44 (95% CI: 1.26, 1.65) while the MF IRR was <1 for all other race and ethnic groups. 

We examined sex differences in cancer incidence by 10-year age group by race and ethnicity (Figure 1 and Appendix A). Pancreas cancer exhibited a female bias among individuals aged 20–29 and a male bias after age 40, a pattern that was consistent across race and ethnicity. Colorectal cancer showed no sex bias in the younger ages, though the male bias steadily increased from ages 40–49 to 70–79. The MF IRR for anus cancer decreased as the population aged and was <1, indicating a female bias among White individuals aged 40 and older, Latino individuals aged 50 and older, and Black individuals aged 60–79 (Figure 1). The MF IRRs for anus cancer among the Black population aged 20–29 and 30–39 were >1, exhibiting a male bias in the earlier years of life. There was an increase in the male bias with age across all race and ethnicities for lung and bronchus cancer, though Black individuals had the highest MF IRR and AIAN individuals had the lowest. Conversely, lung and bronchus cancers exhibited a female bias among White, Hispanic, and AIAN individuals before age 50.

## 4. Discussion

In our analysis of sex differences in cancer incidence, we observed that there were nine cancer sites and two histology types with inconsistent MF IRRs across race and ethnicity, suggesting extrinsic behavioral and environmental factors as potential etiologies for the divergence of MF IRRs between groups. Differences in the MF IRR by race and ethnicity likely suggest differences in lifestyle, behaviors, and environmental factors by gender such as, tobacco use, alcohol consumption, and occupational hazards that contribute to the variation in cancer development. Of the cancer sites with inconsistent MF IRRs by race and ethnicity, cancers of the lip, tongue, hypopharynx, retroperitoneum, larynx, pleura, and Kaposi sarcoma had a male bias, while gallbladder and anus cancers were female biased. Conversely, minimal variation in MF IRRs by race and ethnicity may imply sex differences in genetic factors or immune function that influence cancer susceptibility, regardless of racial or ethnic group. Similarly, most of the MF IRR histologic subtypes for selected cancer sites were consistent across race and ethnicity. Taken together, our results point to intrinsic biological factors as potential explanations for the observed sex difference in cancer incidence due to the low variation of the MF IRRs by race and ethnicity. 

It is possible that there are carcinogenic exposures with a similar sex prevalence by race and ethnicity. For instance, in the U.S., water and air pollution is highest in areas where Black, API, and Latino populations reside, but men and women are likely to be exposed equally [11,12]. Air pollution is associated with lung, bladder, kidney, and colorectal cancers, so we might expect to see very low MF IRRs for these cancers among non-White individuals. Instead, we see consistent MF IRRs across race and ethnicity for these cancers. Alternatively, there are potential carcinogenic exposures that have a similar sex bias across race and ethnicity that results in a consistent MF IRR. However, this is unlikely given the large number of cancers that we observed as having a consistent MF IRR across racial groups. Further, the most common risk factors for cancer, such as cigarette smoking, alcohol use, diet, physical activity, and infectious agents [13], vary greatly by sex and race and ethnicity [14]. 

Previous research that investigated sex differences in cancer incidence, taking into consideration confounding factors, yielded comparable findings. A previous study reported that the majority of non-sex-specific cancers still exhibited MF IRR > 1 after accounting for a plethora of demographic, lifestyle, and dietary confounders [6]. Height might be an additional factor contributing to the explanation of the sex difference in cancer [15,16]. When comparing the crude and height-adjusted cancer risk, it was observed that height mediated the increased risk of kidney cancer, melanoma, and hematologic malignancies in males, though it did not explain the sex differences seen in other cancer types [15,16]. 

Our results show that the MF IRR for anus cancer was <1, indicating a female bias for most racial and ethnic groups, except among Black individuals where a male bias was observed. This observation may be driven by HIV infection. The prevalence of HIV is notably higher in Black and Latino men with anal SCC than White men, and anal SCC cases with HIV have significantly impacted anal SCC incidence trends among men, while not showing the same influence on women without HIV [17]. People with HIV have a higher likelihood of developing anus cancer, with 35% of anal SCCs among males in the general population occurring among people with HIV compared to only 2% of anal SCC among females [17,18]. 

Lung and bronchus cancer was observed to have a relatively consistent MF IRR across race and ethnicity, though this varied by age in our age group analysis. Tobacco smoking increases the risk of developing lung cancer by 15–30 times and is responsible for 80% to 90% of the diagnoses [19]. Consistent with our results, Jemal et al. observed higher rates of lung cancer among young non-Hispanic White and Latina women [20]. This finding is likely not due to sex differences in smoking behaviors as the prevalence of smoking among White women is similar to their male counterparts [20,21]. Further, the smoking prevalence is lower among Latina women than among Latino men across the life course [21]. If sex differences in the prevalence of smoking explained the sex difference in smoking-related cancers, we might expect to see a higher MF IRR for API individuals who have the largest sex difference in smoking prevalence (14% for API men and 4.6% for women) [22]. Instead, the sex ratio for API individuals is similar to that of Black and Latino individuals. 

On the other hand, similarities in MF IRRs across race and ethnicity may be due to intrinsic biologic factors. Genes located on the X-chromosome regulate metabolism, immunity, and tumor suppression, likely providing females with a more robust immune response against infections and cancer [23,24,25]. Sex-related hormones also contribute to cancer initiation and progression by affecting cancer cells, metabolism, and regulation of inflammation [26,27,28]. These potential protective factors in females may account for the observed male bias in many non-sex-specific cancers. Alternatively, for some cancers, sex hormones can also have detrimental effects in cancer development among females. For instance, the female bias in gallbladder and thyroid cancers may be due to increased estrogen exposure [29,30,31,32]. 

This study has several strengths, including large-scale population coverage of the SEER registries and robust cancer ascertainment [33]. However, a limitation of this study is the inability to adjust for important confounders (e.g., smoking, body mass index, and occupation) on an individual level that might contribute to cancer incidence differences between males and females. Had it been possible to account for these confounding variables, we may have observed a higher number of cancers with no sex bias. Another limitation of this study is that the analysis is confined to selected states and regions in the U.S., which may share common environmental and behavioral risk factors. The lack of risk factor data paired with the limited geographic coverage reduces the generalizability of our results to other regions. Additionally, the small sample size of AIAN individuals in this analysis is due to the restriction to PRCDA counties and small size of the population, which limited statistical power and our ability to draw strong conclusions about sex differences in this population. Similarly, due to the small sample size of Pacific Islander individuals, the API group was composed primarily of Asian Americans; therefore, our findings may not generally apply to this group.

## 5. Conclusions

In conclusion, we observed that most non-sex-specific cancers have a higher incidence among males than females, with minimal variation between racial and ethnic groups. Our findings suggest that for most cancer sites, intrinsic biological differences could be driving the sex difference in cancer incidence. Few of the sex differences varied by race and ethnicity, which indicates that environmental exposure differences are unlikely to drive most sex differences in cancer incidence. Future research should investigate the biological underpinnings that are driving sex differences in cancer incidence.

## Figures and Tables

**Figure 1 cancers-16-00989-f001:**
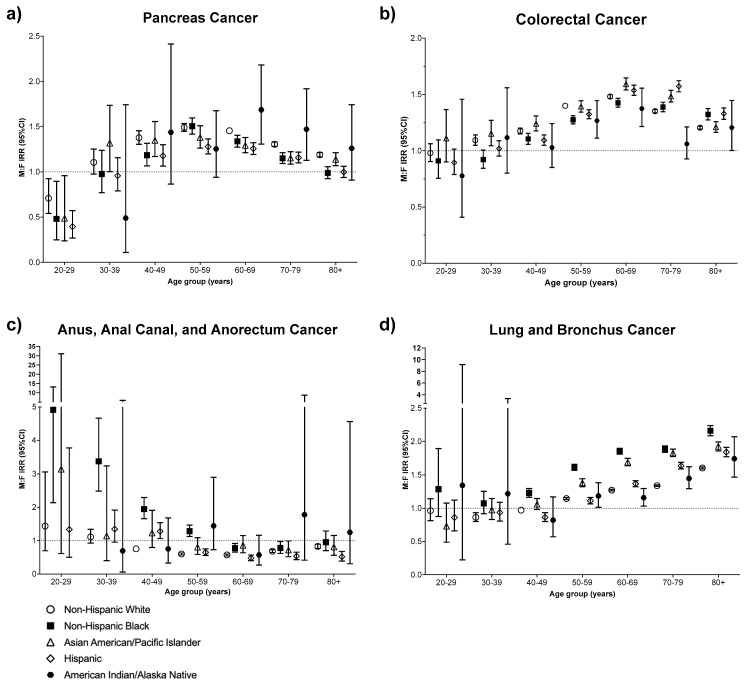
Male-to-female incidence ratios by age and race and ethnic groups. (**a**) Pancreas cancer; (**b**) colorectal cancer; (**c**) anus, anal canal, and anorectum cancer; (**d**) lung and bronchus cancer.

**Table 1 cancers-16-00989-t001:** Cancer site male-to-female incidence rate ratio by racial and ethnic group adjusted for age, SEER 2000–2019.

	Non-Hispanic White		Non-Hispanic Black		Non-Hispanic Asian Pacific Islander		Latino		Non-Hispanic American Indian/Alaskan Native	
	MF IRR	95% CI	MF IRR	95% CI	MF IRR	95% CI	MF IRR	95% CI	MF IRR	95% CI
Lip	3.45	(3.30, 3.61)	2.62	(1.82, 3.80)	1.43	(1.05, 1.96)	2.90	(2.46, 3.44)	3.03	(1.36, 7.28)
Tongue	2.80	(2.74, 2.86)	3.19	(2.96, 3.44)	1.74	(1.62, 1.87)	2.15	(2.01, 2.31)	2.78	(2.00, 3.89)
Salivary Gland	1.93	(1.86, 1.99)	1.17	(1.06, 1.29)	1.19	(1.08, 1.31)	1.31	(1.19, 1.44)	1.17	(0.75, 1.80)
Floor of Mouth	2.41	(2.29, 2.54)	3.71	(3.19, 4.34)	2.06	(1.64, 2.60)	2.69	(2.24, 3.24)	2.30	(1.29, 4.27)
Gum and Other Mouth	1.41	(1.37, 1.45)	1.59	(1.46, 1.73)	1.78	(1.62, 1.95)	1.49	(1.36, 1.63)	1.59	(1.07, 2.39)
Nasopharynx	2.59	(2.42, 2.77)	2.96	(2.59, 3.39)	2.78	(2.59, 2.97)	2.66	(2.28, 3.10)	2.15	(1.47, 3.18)
Tonsil	5.03	(4.87, 5.20)	5.12	(4.65, 5.64)	4.30	(3.69, 5.03)	4.77	(4.30, 5.30)	5.50	(3.50, 9.00)
Oropharynx	3.84	(3.61, 4.10)	4.00	(3.45, 4.66)	3.14	(2.32, 4.28)	4.64	(3.76, 5.76)	4.22	(1.85, 10.97)
Hypopharynx	4.03	(3.81, 4.26)	5.88	(5.14, 6.74)	6.17	(4.97, 7.71)	8.18	(6.68, 10.1)	5.56	(2.98, 11.03)
Other Oral Cavity and Pharynx	3.25	(2.99, 3.53)	3.55	(2.85, 4.44)	3.53	(2.30, 5.53)	3.58	(2.65, 4.90)	2.26	(0.81, 7.60)
Esophagus	4.43	(4.34, 4.53)	3.11	(2.96, 3.28)	3.60	(3.34, 3.89)	4.57	(4.27, 4.89)	3.16	(2.50, 4.03)
Stomach	2.19	(2.16, 2.23)	1.84	(1.78, 1.90)	1.71	(1.66, 1.76)	1.60	(1.55, 1.64)	1.87	(1.62, 2.15)
Small Intestine	1.38	(1.35, 1.42)	1.32	(1.25, 1.39)	1.58	(1.44, 1.74)	1.28	(1.19, 1.37)	1.44	(0.99, 2.11)
Colon and Rectum	1.32	(1.31, 1.33)	1.32	(1.30, 1.34)	1.39	(1.37, 1.41)	1.40	(1.38, 1.42)	1.18	(1.10, 1.26)
Anus, Anal Canal, and Anorectum	0.67	(0.66, 0.69)	1.24	(1.15, 1.33)	0.85	(0.74, 0.98)	0.63	(0.58, 0.69)	0.98	(0.67, 1.42)
Liver	3.47	(3.41, 3.54)	3.75	(3.61, 3.90)	2.94	(2.85, 3.04)	2.84	(2.76, 2.93)	2.53	(2.23, 2.88)
Gallbladder	0.62	(0.60, 0.65)	0.66	(0.60, 0.73)	0.71	(0.64, 0.78)	0.46	(0.43, 0.50)	0.64	(0.47, 0.86)
Other Biliary	1.57	(1.52, 1.62)	1.36	(1.25, 1.48)	1.54	(1.44, 1.64)	1.34	(1.26, 1.42)	1.14	(0.84, 1.54)
Pancreas	1.33	(1.31, 1.34)	1.19	(1.16, 1.23)	1.21	(1.17, 1.25)	1.14	(1.11, 1.17)	1.40	(1.22, 1.61)
Retroperitoneum	1.27	(1.19, 1.35)	0.78	(0.65, 0.93)	0.98	(0.82, 1.17)	1.14	(0.99, 1.32)	0.54	(0.22, 1.27)
Peritoneum, Omentum, and Mesentery	0.08	(0.07, 0.09)	0.21	(0.16, 0.27)	0.06	(0.04, 0.09)	0.12	(0.09, 0.15)	0.11	(0.02, 0.41)
Other Digestive Organs	1.35	(1.28, 1.41)	1.29	(1.14, 1.46)	1.36	(1.18, 1.55)	1.17	(1.05, 1.31)	1.25	(0.81, 1.93)
Nose, Nasal Cavity, and Middle Ear	1.77	(1.69, 1.85)	1.93	(1.69, 2.21)	1.81	(1.58, 2.07)	1.63	(1.45, 1.83)	1.37	(0.77, 2.48)
Larynx	4.38	(4.27, 4.49)	5.64	(5.32, 5.99)	8.68	(7.54, 10.03)	7.69	(7.05, 8.39)	4.38	(3.13, 6.23)
Lung and Bronchus	1.32	(1.32, 1.33)	1.83	(1.81, 1.86)	1.71	(1.68, 1.74)	1.53	(1.51, 1.56)	1.37	(1.28, 1.47)
Pleura	2.23	(1.79, 2.79)	0.91	(0.37, 2.07)	0.76	(0.27, 2.01)	5.23	(2.63, 11.55)	4.38	(0.28, 230.79)
Trachea, Mediastinum, and Other Respiratory Organs	2.62	(2.35, 2.93)	2.29	(1.71, 3.08)	2.57	(1.92, 3.46)	2.63	(2.06, 3.39)	~	~
Bones and Joints	1.36	(1.30, 1.43)	1.28	(1.12, 1.46)	1.30	(1.13, 1.51)	1.29	(1.16, 1.42)	1.34	(0.77, 2.29)
Soft Tissue Including Heart	1.56	(1.53, 1.60)	1.21	(1.15, 1.28)	1.38	(1.30, 1.48)	1.28	(1.21, 1.34)	1.70	(1.29, 2.25)
Skin Excluding Basal and Squamous	1.60	(1.59, 1.62)	1.19	(1.11, 1.28)	1.32	(1.24, 1.41)	1.06	(1.02, 1.10)	1.23	(1.05, 1.45)
Melanoma of the Skin	1.57	(1.56, 1.58)	1.25	(1.12, 1.39)	1.26	(1.16, 1.37)	1.03	(0.99, 1.07)	1.25	(1.05, 1.49)
Other Non-Epithelial Skin	2.14	(2.08, 2.19)	1.14	(1.04, 1.26)	1.42	(1.28, 1.59)	1.20	(1.10, 1.31)	1.17	(0.76, 1.79)
Urinary Bladder	4.15	(4.11, 4.18)	3.00	(2.91, 3.11)	4.17	(4.00, 4.34)	3.86	(3.74, 3.99)	4.14	(3.49, 4.92)
Kidney and Renal Pelvis	2.06	(2.04, 2.08)	1.99	(1.94, 2.04)	2.11	(2.04, 2.19)	1.81	(1.77, 1.85)	1.87	(1.7, 2.06)
Ureter	2.17	(2.06, 2.27)	1.72	(1.39, 2.15)	1.68	(1.46, 1.95)	1.90	(1.60, 2.26)	1.69	(0.62, 5.17)
Other Urinary Organs	3.31	(3.09, 3.55)	1.76	(1.49, 2.07)	2.11	(1.68, 2.67)	2.63	(2.14, 3.24)	4.11	(1.40, 15.14)
Eye and Orbit	1.41	(1.35, 1.46)	1.62	(1.21, 2.17)	1.54	(1.21, 1.97)	1.44	(1.26, 1.65)	1.20	(0.55, 2.54)
Brain	1.49	(1.47, 1.51)	1.48	(1.39, 1.57)	1.49	(1.41, 1.58)	1.36	(1.31, 1.42)	1.50	(1.19, 1.91)
Cranial Nerves Other Nervous System	1.02	(0.95, 1.09)	1.05	(0.87, 1.27)	1.05	(0.86, 1.28)	1.02	(0.87, 1.19)	0.84	(0.31, 2.07)
Thyroid	0.37	(0.37, 0.37)	0.30	(0.28, 0.31)	0.32	(0.31, 0.33)	0.28	(0.27, 0.29)	0.29	(0.25, 0.35)
Other Endocrine Including Thymus	1.25	(1.18, 1.31)	1.02	(0.91, 1.15)	1.32	(1.17, 1.47)	1.18	(1.05, 1.34)	1.49	(0.71, 3.20)
Hodgkin Lymphoma	1.27	(1.24, 1.30)	1.34	(1.25, 1.42)	1.33	(1.22, 1.47)	1.40	(1.33, 1.48)	1.13	(0.76, 1.68)
Non-Hodgkin Lymphoma	1.47	(1.45, 1.48)	1.47	(1.43, 1.51)	1.46	(1.42, 1.50)	1.32	(1.30, 1.35)	1.26	(1.11, 1.42)
Myeloma	1.67	(1.65, 1.70)	1.39	(1.35, 1.43)	1.47	(1.39, 1.55)	1.44	(1.39, 1.49)	1.19	(0.98, 1.43)
Lymphocytic Leukemia	1.92	(1.89, 1.95)	2.00	(1.90, 2.11)	1.76	(1.64, 1.90)	1.56	(1.49, 1.63)	1.87	(1.44, 2.44)
Myeloid and Monocytic Leukemia	1.58	(1.55, 1.60)	1.40	(1.34, 1.47)	1.55	(1.48, 1.62)	1.44	(1.38, 1.49)	1.38	(1.12, 1.71)
Other Leukemia	1.43	(1.37, 1.50)	1.28	(1.13, 1.44)	1.69	(1.45, 1.98)	1.43	(1.26, 1.62)	1.62	(0.98, 2.69)
Mesothelioma	4.18	(4.01, 4.36)	4.25	(3.61, 5.03)	3.35	(2.76, 4.08)	3.44	(3.09, 3.85)	3.20	(1.70, 6.29)
Kaposi Sarcoma	12.86	(11.58, 14.32)	13.5	(11.43, 16.06)	13.81	(9.08, 22.00)	7.96	(6.78, 9.40)	6.33	(2.57, 20.83)

Abbreviations: MF IRR, male-to-female incidence rate ratio; CI, confidence interval. ~ Statistic could not be calculated. Age-adjusted to the 2000 U.S. std population (19 age groups—Census P25-1130) standard [10]; confidence intervals (Tiwari mod) are 95% for ratio.

**Table 2 cancers-16-00989-t002:** Male-to-female incidence rate ratio of histology of cancer subtypes by racial and ethnic group adjusted for age, SEER 2021, 2000–2019.

	Non-Hispanic White		Non-Hispanic Black		Latino		Non-Hispanic Asian Pacific Islander		Non-Hispanic American Indian/Alaskan Native	
	MF IRR	95% CI	MF IRR	95% CI	MF IRR	95% CI	MF IRR	95% CI	MF IRR	95% CI
Esophagus Adenocarcinoma	7.71	(7.48, 7.95)	3.80	(3.30, 4.38)	6.10	(5.08, 7.37)	6.10	(5.51, 6.76)	4.07	(2.85, 5.95)
Esophagus SCC	1.75	(1.69, 1.81)	2.98	(2.81, 3.16)	3.13	(2.86, 3.43)	3.46	(3.12, 3.84)	2.87	(1.98, 4.20)
Gastric Cardia Adenocarcinoma	4.88	(4.72, 5.04)	3.05	(2.72, 3.41)	3.82	(3.44, 4.26)	2.98	(2.74, 3.24)	3.10	(2.15, 4.53)
Gastric Non-Cardia Adenocarcinoma	1.62	(1.58, 1.66)	1.95	(1.87, 2.03)	1.61	(1.56, 1.67)	1.59	(1.54, 1.65)	1.90	(1.60, 2.26)
Liver HCC	4.07	(3.99, 4.16)	4.17	(4.00, 4.36)	3.13	(3.03, 3.24)	3.11	(3.01, 3.21)	2.70	(2.35, 3.10)
Liver ICC	1.26	(1.21, 1.31)	1.36	(1.20, 1.53)	1.35	(1.23, 1.47)	1.08	(0.99, 1.17)	1.29	(0.85, 1.95)
Lung and Bronchus Adenocarcinoma	1.09	(1.08, 1.09)	1.48	(1.45, 1.51)	1.26	(1.23, 1.29)	1.22	(1.19, 1.25)	1.06	(0.94, 1.21)
Lung and Bronchus SCC	2.01	(1.99, 2.04)	2.59	(2.51, 2.66)	3.46	(3.29, 3.64)	2.36	(2.26, 2.47)	1.84	(1.60, 2.12)
Lung and Bronchus Small Cell Carcinoma	0.79	(0.76, 0.81)	1.09	(0.99, 1.19)	1.44	(1.26, 1.65)	0.83	(0.76, 0.91)	0.96	(0.59, 1.54)
Lung and Bronchus Large Cell Carcinoma	1.72	(1.25, 2.37)	1.39	(0.54, 3.60)	17.76	(2.48, 753.32)	3.89	(1.13, 17.56)	~	~
Urinary Bladder TCC	4.28	(4.24, 4.32)	3.20	(3.09, 3.32)	4.35	(4.17, 4.53)	4.08	(3.95, 4.23)	4.86	(4.01, 5.92)

Abbreviations: MF IRR, male-to-female incidence rate ratio; CI, confidence interval; SCC, squamous cell carcinoma; HCC, hepatocellular carcinoma; ICC, intrahepatic cholangiocarcinoma; TCC, transitional cell carcinoma. ~ Statistic could not be calculated. Age-adjusted to the 2000 U.S. std population (19 age groups—Census P25-1130) standard [10]; confidence intervals (Tiwari mod) are 95% for ratios.

## Data Availability

The data that support the findings of this study are publicly available from https://seer.cancer.gov/ (accessed on 18 August 2023).

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
