# Peer review of "Sex Differences in Cancer Incidence Rates by Race and Ethnicity: Results from the Surveillance, Epidemiology, and End Results (SEER) Registry (2000–2019)"

_cancers, 2024, doi:10.3390/cancers16050989_

Round 1

Reviewer 1 Report

Comments and Suggestions for Authors

This manuscript presents analyses of the sex ratio of male to female incidence rates for a large number of cancer sites and subtypes by race and ethnicity.  The aim is to increase understanding of whether sex differences in incidence are driven by intrinsic biological differences or by differences in environmental and behavioural factors by comparing sex ratios overall and by age group between populations of different race and ethnicity.  The data are from the high-quality SEER Program of the United States.  The analytic methods are appropriate, the results are well presented, and the conclusions are justified by the data.  The manuscript would benefit from more detailed description of the dataset and some additional analysis, as suggested in the following specific comments.

1. While the SEER Program is justly well-known, the interested reader would be saved the trouble of referring to other sources if there were a (probably supplemental) table listing the contributing cancer registries and their total populations at risk, either for a representative year (e.g. 2010) or averaged over the 20-year study period.  A sentence in the text could usefully state the percentage of the US population covered by SEER.

2. It would also be useful to have a table (again maybe supplemental) with the definitions of all cancer sites and selected histological subtypes according to ICD-O-3 codes.  If such a table is provided, then the second paragraph of Materials and Methods would be redundant since all the information would then be included in the table.

3. The Results should include a straightforward 3-part, 1-dimensional table of the total numbers of cases by cancer site and subtype, the total numbers by racial/ethnic group, and the total numbers by 10-year age group.

4. The numbers of cases for some of the cancers studied must be very large.  A valuable addition to the paper would be a comparison of results for the two halves of the study period, 2000-2009 and 2010-2019.  If any marked differences emerge, these could be discussed in terms of changes over time in environmental and behavioural factors, and perhaps also changes in the racial composition of the very heterogeneous API and Latino populations.  Results might be especially interesting for cancers where IRR varies with age.  For instance, if the age pattern changed between the two periods, this could be indicative of a cohort effect.

Two minor points.

5. Lines 58-59.  This is the only time that Mexican-Americans are mentioned.  The relevance of the sentence is unclear without knowing (i) the proportion of the Latino population that is Mexican-American and (ii) whether the pattern of obesity among Mexican-Americans is known, or at least believed, to be typical of the Latino population as a whole.

6. Line 148.  The Supplemental Figure was absent from my copy of the manuscript, so I am unable to comment on it.  What does it show?  Does it still exist in the current version?

Reviewer 2 Report

Comments and Suggestions for Authors

Authors AAAuthors investigated + sex differences in cancer incidence by race and ethnicity using the cancer incidence data from the Surveillance, Epidemiology and End Result (SEER) program (2000-2019). Among 49 cancer sites, 44 showed male predominance (MF IRR >1), with seven inconsistencies across race and ethnicity, including cancers of the lip, tongue, hypopharynx, retroperitoneum, larynx, pleura cancers, and Kaposi sarcoma. Four cancers exhibited a female predominance (MF IRR <1), with only gallbladder and anus cancers varying by race and ethnicity.

This is a really interesting epidemiological study.

Can you please explain shortly how you calculated IRRs and 95% CI? What analysis method did you use? Regression models, which? 

What software was used (SAS; SPSS; STRATA; or what other)?

Reviewer 3 Report

Comments and Suggestions for Authors

It is a really interesting study, investigating sex differences in incidence of a variety of common cancers. A major limitation, already acknowledged by authors, is that they did not analyzed the impact of well-known risk factors.

Major comments:

Authors should clarify the proportion of cancer cases that is registered to the SEER database as well as the total number of patients included in their analysis and number of patients at each subgroup of interest.

Authors should add at the study limitations that their analysis is restricted to US population. Although they analyzed MF IRR by race and ethnicity, the geographical area is rather restricted, meaning that the study population may share environmental and maybe behavioral risk factors. Although this fact supports the results of the present study in regard to effect of sex, as risk factors were not analyzed, these results may not be applicable to other geographical areas.

Round 2

Reviewer 1 Report

Comments and Suggestions for Authors

All review comments satisfactorily addressed.